# Elevation of the Blood Glucose Level is Involved in an Increase in Expression of Sweet Taste Receptors in Taste Buds of Rat Circumvallate Papillae

**DOI:** 10.3390/nu12040990

**Published:** 2020-04-02

**Authors:** Moemi Iwamura, Risa Honda, Kazuki Nagasawa

**Affiliations:** Department of Environmental Biochemistry, Division of Biological Sciences, Kyoto Pharmaceutical University, Kyoto 607-8414, Japan; ky14049@ms.kyoto-phu.ac.jp (M.I.); ky14323@ms.kyoto-phu.ac.jp (R.H.)

**Keywords:** sweet taste, gustation, taste bud, blood glucose level, diabetes mellites, fasting and fed condition

## Abstract

The gustation system for sweeteners is well-known to be regulated by nutritional and metabolic conditions, but there is no or little information on the underlying mechanism. Here, we examined whether elevation of the blood glucose level was involved in alteration of the expression of sweet taste receptors in circumvallate papillae (CP) and sweet taste sensitivity in male Sprague-Dawley rats. Rats under 4 h-fed conditions following 18 h-fasting exhibited elevated blood glucose levels and decreased pancreatic T1R3 expression, compared to rats after 18 h-fasting treatment, and they exhibited increased protein expression of sweet taste receptors T1R2 and T1R3 in CP. Under streptozotocin (STZ)-induced diabetes mellites (DM) conditions, the protein expression levels of T1R2 and T1R3 in CP were higher than those under control conditions, and these DM rats exhibited increased lick ratios in a low sucrose concentration range in a brief access test with a mixture of sucrose and quinine hydrochloride (QHCl). These findings indicate that the elevation of blood glucose level is a regulator for an increase in sweet taste receptor protein expression in rat CP, and such alteration in STZ-induced DM rats is involved in enhancement of their sweet taste sensitivity.

## 1. Introduction

Mono- and di-saccharides such as sucrose, glucose, fructose, etc., are macronutrients and induce sweet taste sensation as a preferable energy source by stimulating sweet taste receptors expressed by taste cells in taste buds of the tongue [1,2]. Sweet taste receptors consist of a heterodimer of T1R2 and T1R3 or a homodimer of T1R3, and are functionally expressed in a variety of organs such as the gut, pancreas, respiratory tract, brain, etc., in addition to the tongue [3]. For instance, in the gut, activation of T1R3 expressed by gut mucosa cells induces the release of hormones such as glucagon-like peptide-1 into the blood circulation, and up-regulates expression of the glucose transporter SGLT1 in the brush border membranes for efficient absorption of monosaccharides as an energy source [3]. In the pancreas, sweet taste receptors play a role in sensing the blood glucose level to regulate insulin secretion from ß cells [4,5,6,7]. Judging from these findings, sweet taste receptors are necessary for fine tuning of glucose homeostasis in the body.

Recent accumulating evidence clearly indicates that nutritional and metabolic conditions play roles in the regulation of sweet taste sensitivity. As the underlying mechanisms, modulation of the functionality of sweet taste receptors, signal transduction between taste cells in taste buds, neural transmission, and/or perception by the central nervous system are considered to be involved in. In the study by Chen et al. [8], rats under fasting conditions exhibited decreased gustatory thresholds for sweet taste, while rats with diabetes mellites (DM) induced by streptozotocin (STZ) administration and obesity induced by a high-fat diet fed exhibited an enhanced response of chorda tympani nerves to sweeteners. Under hunger and satiety conditions, on the other hand, the primary taste cortex responsible for taste perception [9] and taste nerve signals [10] are reported to play no or only negligible roles in alteration of taste sensitivity, while taste buds are suggested to serve as an important target for gustatory sensitivity modulation [11]. In addition, the sweet taste sensitivity of taste cells appears to be regulated by satiety-related hormones such as leptin [3 and references therein]. Therefore, regulation of the functional expression of the sweet taste perception system in the tongue for immediate control of the sensitivity to sweet taste according to nutritional demands is reasonable.

It has been demonstrated that the expression level of α-gustducin, a downstream molecule in T1R2/T1R3-mediated sweet taste signaling, in taste buds is correlated with the response to sweet taste in rats under fasting and fed conditions [8], and in DM rats [12]. In addition, a decreased recognition threshold for sweet taste was detected in human subjects under fasting conditions compared with after a meal [13]. Since the blood glucose level shows fluctuation under fasting and fed conditions, there is a possibility that functional alteration of the sweet taste sensing system, especially taste receptors, might be induced by alteration in the blood glucose level.

In this study, therefore, we examined whether or not the blood glucose level was involved in regulation of functional expression of sweet taste receptors in the taste buds of rats under fasting, fed and STZ-induced DM conditions. In addition, we evaluated sweet taste sensitivity of STZ-induced DM rats by a brief access test with a mixture of sucrose and QHCl. Based on the finding obtained here, we concluded that the blood glucose level is a regulator of sweet taste sensitivity by controlling the sweet taste receptor expression in taste buds.

## 2. Materials and Methods

### 2.1. Materials

STZ, citric acid, sucrose and QHCl were purchased from Fujifilm Wako Pure Chemical Corporation (Osaka, Japan), collagenase D and dispase II from Roche Diagnostics Ltd. (Mannheim, Germany), and trypsin inhibitor and bovine serum albumin (BSA) from Sigma-Aldrich (St. Louis, MO, USA). The primer sets were generated by Eurofins Genomics K.K. (Tokyo, Japan) and their sequences are shown in Table 1. The antibodies used for Western blotting and immunohistochemistry are shown in Table 2; Table 3, respectively. Except where otherwise noted, the chemicals and reagents were obtained from Fujifilm Wako Pure Chemical Corporation.

### 2.2. Animals

Male Sprague-Dawley rats (6–8 weeks old; Japan SLC, Hamamatsu, Japan) were housed with food and water available ad libitum in a controlled environment with a 12 h/12 h light/dark cycle in the specific pathogen-free facility. All experiments were performed in strict accordance with ARRIVE guidelines, and were approved by the Experimental Animal Research Committee of Kyoto Pharmaceutical University (authorization number: 17-003 from 2017 to 2019). The number of animals was kept to the minimum necessary for a meaningful interpretation of the data, and animal discomfort was minimized. After 1 week habituation to the animal facility, rats were randomly divided into two groups, and to evaluate water intake and food consumption, they were housed individually in a cage.

### 2.3. Fasting and Fed Conditions

Based on the report of Medina et al. [7], before dissection, rats that fasted for 18 h were defined as fasting ones, and the fed ones were allowed free access to a regular unrestricted diet (MF, Oriental Yeast Co. Ltd., Tokyo, Japan) for 4 h following 18 h-fasting. In both groups, rats were allowed to drink water freely during experimental periods.

### 2.4. Development of DM in Rats

In the DM group, rats were administered intraperitoneally (*i.p.*) a dose of 50 mg/kg of STZ, which was dissolved in 0.1 M citric acid buffer (pH 4.5), and ones in the control group were administered *i.p.* 0.1 M citric acid buffer (pH 4.5) as a vehicle [14,15]. STZ is known to induce type 1 DM in rats and mice, because they had no insulin secretion from pancreatic cells [16,17]. To confirm development of DM, blood glucose levels under 2 h-fed conditions following 18 h-fasting were measured 2, 9, 16, 23 and 30 days after the STZ administration using a commercially available kit (Glucocard MyDIA^®^, Arkray, Inc., Kyoto, Japan), and rats with levels of 200 or more mg/dL were defined as DM rats [14]. After an additional 2 h-fed treatment, the tongues of rats were obtained for the biochemical analyses described below.

### 2.5. Brief Access Test

As reported previously [18,19], we performed a brief-access test to evaluate behavioral responses to tastants in rats. Since a brief access test is suitable for testing orosensory responses without a significant post-ingestive contribution, it permits appropriate evaluation of relationship between blood glucose levels and sweet taste sensitivity and receptor expression. All training and test sessions were performed during the light phase of the light/dark cycle. The rats had restricted access to water for 18 h before each training or test session. To obtain a stable lick number, which means motivation of a rat to drink water with or without a tastant, a training session was performed for 8 days before initiation of a test session. On the first day of a training session, so that a rat would get used to the experimental apparatus, it was placed in a test box (a black box to shield from light; width: 36.5 cm, depth: 21.5 cm and height: 25.5 cm), and was given free-access to distilled water for 15 min from a polypropylene tube via an elliptical window (major axis: 15 mm, minor axis: 10 mm). A lick counter system (INECK, Kyoto, Japan) was set up between the edge of the tube and the window, and the lick number were determined automatically by recording the number of interceptions of the sensor beam by the tongue of a rat as it licked the solution from the edge of the tube. On the second day of the training session, the rat was trained to drink distilled water on an interval schedule, consisting of a 10 s-period of presentation of distilled water and a 20 s inter-presentation interval, and this schedule was repeated 15–25 times. On the third day, the training was performed with the same procedure as on the second day except for the use of a 0.3 M sucrose solution instead of distilled water. On the fourth day, the 0.3 M sucrose solution containing QHCl at the concentration of 0.3 mM was presented to the rat for 10 s alternatively with 10 s-presentation of distilled water as an interval between the two taste solutions. The use of a mixture of sweet and bitter taste solutions has been reported to allow sensitive detection of a change in sweet taste sensitivity, because when a rat cannot detect the sweet taste in the mixture, it only perceives the bitter taste, and thus avoids drinking the mixture [20]. On the fifth to seventh days, the training was performed with the same procedure as that for a test session described below, and the lick numbers obtained on the eighth day were used as the base data (day 0). After a 18 h-water deprivation period, rats were allowed to drink distilled water for 2 min freely to quench their excess thirst. Then, each rat was subjected to a test session. In the test session, the lick numbers were determined 3, 10 and 17 days after STZ administration. The test session was designed to detect the concentration-dependent response to sucrose for quantification of sweet taste sensitivity [20]. The taste solutions used in the experiments were prepared as mixtures of a concentration range of sucrose (0.01 to 1 M) and 0.3 mM QHCl. A series of sucrose-QHCl mixtures was presented with sucrose concentrations of descending order in the mixtures. During the test session, the taste solution and distilled water were alternatively presented to the rat for 10 and 5 s, respectively, presentations of distilled water being designed to rinse the oral cavity. Data were expressed as lick ratios as a quantitative index of taste sensitivity, which were calculated by dividing the lick number for a taste solution in 10 s by that for distilled water. This is because the lick number is affected by the difference in motivation to drink solutions among rats. The lick ratio versus sucrose concentration curves were non-linearly fitted using GraphPad Prism Software version 6 (GraphPad Software, Inc., La Jolla, CA, USA). We excluded the data for rats that could not accomplish a series of lick tests on each test day.

### 2.6. Exfoliation of Epithelial Tissue Including Rat Circumvallate Papillae (CP)

Rats were perfused transcardially with saline under deep anesthesia (pentobarbital sodium, 25 mg/kg, *i.p.*). As reported previously, rat lingual epithelial tissues containing CP were exfoliated from the tongue by subcutaneous injection of an enzyme cocktail comprising 2.5 mg/mL dispase II, 1.0 mg/mL collagenase D, and 1.0 mg/mL trypsin inhibitor for 30 min at room temperature, and then the epithelial tissues were treated with an RNAlater^®^ solution (Sigma-Aldrich) and kept at –20 °C until use [21].

### 2.7. Reverse Transcription (RT) and Real-Time Quantitative PCR Analyses

Total RNA was extracted and reverse transcribed with a GenElute^™^ Mammalian Total RNA kit (Sigma-Aldrich) and a PrimeScript^™^ RT reagent kit with gDNA Eraser (Takara, Shiga, Japan), respectively, according to the manufacturers’ instruction manuals. Real-time quantitative PCR was performed with a LightCycler^®^ 96 System (Roche Diagnostics) using a KAPA SYBR FAST qPCR Master Mix (2X) Kit (Kapa Biosystems, Wilmington, MA, USA). PCR amplification was performed using the gene specific primer sets shown in Table 1, and all reactions were carried out with the following cycling parameters: 94 °C for 3 min, and 40 cycles of 95 °C for 10 s, 60 °C for 20 s, and 72 °C for 1 s. mRNA expression levels of target genes were normalized against the corresponding levels of β-actin mRNA.

### 2.8. Western Blotting

Following the method reported previously [22], samples were subjected to sodium dodecyl sulfate-polyacrylamide gel electrophoresis (SDS-PAGE), and then transferred to polyvinylidene fluoride (PVDF) membranes. After blocking, each membrane was incubated at 4 °C overnight with primary antibodies diluted with the blocking buffer (Table 2). After several washes, each membrane was incubated with secondary antibodies diluted with PBS-T (0.1% Tween-20 containing PBS) for 1 h at room temperature (Table 2). The membrane was then washed and the signal was detected with ECL reagent (Perkin Elmer, Inc., Waltham, MA, USA). The optical density of each protein band was determined with Image J software (ver. 1.48; NIH, Bethesda, MD, USA) [23].

### 2.9. Immunohistochemical Analysis

Animals were perfused transcardially with 4% paraformaldehyde in 0.1 M phosphate buffer (pH 7.4) containing 0.2% picric acid under deep anesthesia (pentobarbital sodium, 25 mg/kg, *i.p.*), and then their tongues were removed. The tongues were sectioned at 40 μm thickness with a freezing microtome (Leica CM1850; Leica, Nussloch, Germany), and then the sections were subjected to immunohistochemical analysis [21,24].

Immunoreactivity of antigens was investigated by free-floating immunohistochemistry. Free-floating sections were immunoreacted with primary antibodies (Table 3) in PBS containing 1% donkey serum, 0.3% Triton-X-100, 0.3% BSA and 0.05% sodium azide for 3 days at 4 °C, followed by incubation for a day at 4 °C with secondary antibodies (Table 3) in the same buffer as that for the primary antibodies and 1 mg/mL Hoechst 33258 in the blocking buffer for counterstaining of nuclei. For all immunostaining, a negative control (NC), which was prepared by omitting the primary antibodies, was prepared, and the reproducibility of immunostaining was confirmed by assessing sections from four or five rats per immunostaining. The sections were mounted on glass slides and then enclosed using VECTASHIELD^®^ Mounting Medium (Vector Laboratories, Burlingame, CA, USA). Photomicrographs were obtained under a confocal laser microscope (LSM800; Carl Zeiss, Jena, Germany). Fluorescence intensity was measured using the histogram program of the Photoshop software (Adobe Systems, San Jose, CA, USA) [18,19].

### 2.10. Statistical Analysis

All data are expressed as means ± SD. To detect significant differences in biochemical parameters and behavioral tests between the groups, Student’s *t*-test (Figure 1 and Figure 2, 4c and 5b) and two-way repeated measures ANOVA followed by Tukey’s post hoc multiple comparisons test (Figure 3 and Figure 4a), respectively, were performed. A *p*-value of 0.05 or less was considered statistically significant.

## 3. Results

### 3.1. Sweet Taste Receptor Expression Under Fasting and Fed Conditions

First, in order to confirm whether or not our fed treatment following fasting of rats induced the elevated blood glucose levels and increased expression of pancreatic T1R3 as the report of Medina et al. [7], we evaluated the blood glucose levels and expression of T1R3 in the pancreas of rats under fasting and fed conditions. The blood glucose levels after the 4 h-fed treatment following 18 h-fasting were significantly greater than those after 18 h-fasting only treatment (Figure 1a, *t* = 17.8, *p* < 0.001). The expression levels of T1R3 in the pancreas of these rats were determined by Western blotting (Figure 1b,c). The fed rats exhibited lower expression levels of T1R3 in the pancreas compared to in fasting rats (*t* = 3.34, *p* = 0.0287). These results were almost identical to those reported by Medina et al. [7], indicating the validity of our experimental setting.

Figure 2 shows the expression of sweet taste receptors T1R2 and T1R3 in CP of fasting and fed rats. Although there was no significant alteration in the expression levels of mRNAs for sweet taste receptors (T1R2: *t* = 0.441, *p* = 0.675; T1R3: *t* = 0.780, *p* = 0.465) (Figure 2a), the protein expression levels of both T1R2 and T1R3 were greater in the fed rats than in the fasting ones (T1R2: *t* = 2.31, *p* = 0.0158; T1R3: *t* = 2.57, *p* = 0.00817) (Figure 2b,c), and were significantly correlated with their blood glucose levels (T1R2: Pearson’s correlation coefficient (*r*) = 0.678, *t* = 2.61, *p* = 0.0261; T1R3: *r* = 0.772, *t* = 3.43, *p* = 0.00643) (Figure 2d,e).

### 3.2. Sweet Taste Receptor Expression Under STZ-Induced DM Conditions

The blood glucose level, body weight change and intake of food and water in the control and STZ-administered rats are shown in Figure 3. Administration of STZ apparently increased the blood glucose level from 2 to 31 days after the administration (Figure 3a). As for the blood glucose level, two-way ANOVA revealed a significant main effect of treatment (*F*_(1, 14)_ = 1799, *p* < 0.001), a significant main effect of time (*F*_(5, 70)_ = 75.0, *p* < 0.001), and a significant interaction between treatment and time (*F*_(5, 70)_ = 81.0, *p* < 0.001). The STZ-administration decreased the body weight gain (a significant main effect of treatment (*F*_(1, 14)_ = 198, *p* < 0.001), a significant main effect of time (*F*_(5, 70)_ = 83.8, *p* < 0.001), and a significant interaction between treatment and time (*F*_(5, 70)_ = 20.9, *p* < 0.001)) (Figure 3b), increased the water intake (a significant main effect of treatment (*F*_(1, 14)_ = 778, *p* < 0.001), a significant main effect of time (*F*_(5, 70)_ = 67.9, *p* < 0.001), and a significant interaction between treatment and time (*F*_(5, 70)_ = 44.0, *p* < 0.001)) (Figure 3c), and food consumption (a significant main effect of treatment (*F*_(1, 14)_ = 91.6, *p* < 0.001), a significant main effect of time (*F*_(5, 70)_ = 118, *p* < 0.001), and a significant interaction between treatment and time (*F*_(5, 70)_ = 17.6, *p* < 0.001)) (Figure 3d) in rats. These results clearly indicated that STZ administration induces DM conditions in rats.

Next, we assessed whether sweet taste sensitivity was altered in STZ-induced DM rats. As shown in Figure 4a, in a low concentration range (from 0 to 0.3 M) of sucrose, the lick ratios were significantly increased in STZ-induced DM rats on days 3, 10 and 17 compared to in control ones. Regarding the increase of lick ratios, two-way ANOVA indicated a significant main effect of treatment (*F*_(1, 14)_ = 7.83, *p* = 0.0062), a significant main effect of time (*F*_(5, 70)_ = 15.5, *p* < 0.001), and no significant interaction between treatment and time (*F*_(5, 70)_ = 1.07, *p* = 0.387). In these rats, we found the significant correlation between blood glucose levels and lick ratios at 30 mM sucrose, at which the lick ratios in STZ-induced DM rats were increased time-dependently from day 0 to day 17 (*r* = 0.485, *t* = 2.60, *p* = 0.0163) (Figure 4b). Under STZ-induced DM conditions, thus, the sweet taste sensitivity of rats might be increased, but in the brief access test, we used a mixture of sucrose and QHCl as described in the Materials and Methods, there being a possibility that a decrease in the bitter taste sensitivity of rats under the DM conditions resulted in an increase in the lick ratio.

In these rats, thus, we determined the expression levels of mRNAs for sweet and bitter taste receptors in their CP. As shown in Figure 4c, there was no significant alteration in mRNA expression of sweet taste receptors in the rat CP between the two groups (T1R2: *t* = 2.45, *p* = 0.806; T1R3: *t* = 2.45, *p* = 0.120), while the expression levels of mRNAs for T2R2, T2R10 and T2R38 tended to decrease in STZ-induced DM rats (T2R7: *t* = 2.45, *p* = 0.0974; T2R10: *t* = 2.45, *p* = 0.0871; T2R38: *t* = 2.45, *p* = 0.0978).

Because the expression of sweet taste receptors was altered at the protein, but not mRNA, levels in rats under fed and fasting conditions (Figure 2), we measured expression of T1R2 and T1R3 in CP of STZ-induced DM rats in another experimental set, in which STZ-administered rats exhibited greater blood glucose levels, water intake and food consumption, and less body weight gain as in the case shown in Figure 3 (data not shown). In this study, we could not evaluate protein expression of bitter taste receptors in rat CP, because there were no available antibodies for their immunohistochemical detection. As shown in Figure 5a, the immunofluorescent intensities of T1R2 and T1R3 in CP of STZ-induced DM rats were greater than those in control ones, and their quantification demonstrated a significant increase in their protein expression (T1R2: *t* = 2.45, *p* = 0.00105; T1R3: *t* = 2.45, *p* = 0.00938) (Figure 5b). The expression levels of T1R2 and T1R3 in these rats were correlated with their blood glucose levels (T1R2: *r* = 0.876, *t* = 4.45, *p* = 0.00214; T1R3: *r* = 0.874, *t* = 4.41, *p* = 0.00225) (Figure 5c), and the levels of T1R2 were correlated with their lick ratios (*r* = 0.821, *t* = 3.53, *p* = 0.00778), although the levels of T1R3 exhibited tendency of correlation with the lick ratios (*r* = 0.623, *t* = 1.95, *p* = 0.0866) (Figure 5d). Therefore, under STZ-induced DM conditions, rats might exhibit increased sweet and decreased bitter taste sensitivity, and their increased sweet taste sensitivity was due, at least in part, to the increased expression of sweet taste receptors.

## 4. Discussion

Here, we found that (1) under fed conditions, rats exhibited greater levels of blood glucose and expression of T1R2 and T1R3 in taste buds of CP than those under fasting ones, (2) in STZ-induced DM rats, the lick ratios in the brief access test were increased in the low sucrose concentration range, and the expression levels of T1R2 and T1R3 in taste buds of their CP were increased compared with in control rats, and (3) the expression levels of sweet taste receptors in fed and fasting rats, and STZ-induced DM ones were correlated with their blood glucose levels. These findings suggest that the blood glucose level is a regulator of the expression levels of sweet taste receptors in the taste buds of rat CP and their sweet taste sensitivity.

In this study, we revealed that the expression levels of sweet taste receptors in taste buds of CP are up-regulated, at least in part, by elevated blood glucose levels, which were induced not only in fed rats following fasting treatment, but also by hyperglycemia in STZ-treated rats. Because elevation of the blood glucose level under the 4 h-fed conditions following 18 h-fasting caused an increase in sweet taste receptor expression in CP, this alteration is considered to be an immediate response, and this is supported by the finding of no alteration of mRNA levels for T1R2 and T1R3. This was almost identical to the finding of Zverev where the recognition threshold for sucrose in human subjects was 1 h higher after a meal than during 14–16 h fasting [13]. Kyriazis et al. [6] suggested that the expression and function of sweet taste receptors in pancreatic ß cells, which play a role in sensing plasma glucose levels during fasting to decrease insulin secretion, are reduced under elevated fasting glucose conditions, leading to basal insulin hypersecretion, and this alteration of functional expression of sweet taste receptors disappears during immediate fasting. These findings imply that sweet taste receptors expressed by ß cells play an important role in maintaining the ambient glucose level in a narrow physiological range by sensing small changes in it specifically around distinct short-term fasting glucose levels often seen between meals. In addition, Mace et al. [25] showed that the protein expression of T1R3 in brush border membranes of the intestinal mucosa was reduced by short time treatment with high concentrations of glucose. Together, our findings in this study have suggested the possibility that the increased expression of sweet taste receptors in CP in immediate response to elevation of blood glucose levels might enhance food intake to obtain more energy sources by increasing sweet taste sensitivity.

The increase in expression of sweet taste receptors was a rather immediate response under fed conditions following fasting treatment, and this was considered to be due to post-transcriptional regulation of their protein, but not gene expression and satiety-related hormonal regulation such as leptin [26,27] and insulin [28]. In mouse pancreatic islets, a rapid decrease of T1R3 under fed conditions is suggested to be induced by an increase in degradation of T1R3 protein [7]. A similar scenario seems to be adopted to regulation of taste receptor expression in taste buds of CP. In STZ-induced type 1 DM rats, there is a possibility that this post-transcriptional regulation of sweet taste receptor expression might become chronic, but further investigations are necessary to obtain definitive evidence on immediate and chronic regulatory mechanisms for taste receptor expression in taste buds of CP.

In addition to altered sweet taste sensitivity, STZ-induced type 1 DM rats are considered to have decreased bitter taste sensitivity, because the rats exhibited increased lick ratios with the 0 M sucrose solution, which was a bitter solution consisting of 0.3 mM QHCl in the brief access test with a tendency of a decrease of mRNA expression for some bitter taste receptors in their CP. This decreased bitter taste sensitivity might also be observed only in STZ-induced type 1 DM rats, but bitter taste sensitivity plays an important role in the judgement of whether foods ingested are suitable for consumption or should be rejected. Moreover, bitter taste sensitivity is correlated positively with the percentage of food dislike [29], and a bitter taste receptor haplotype is associated with altered glucose and insulin homeostasis [30]. Thus, further detail investigations on bitter taste sensitivity in DM patients are needed.

Based on the findings obtained in this study we suggest that taste sensitivity and expression of taste receptors, especially for sweet taste, should be evaluated under conditions taking the blood glucose level into consideration.

## 5. Conclusions

We demonstrate that elevation of the blood glucose level in rats under fed conditions following fasting treatment increases the expression levels of sweet taste receptors in their taste buds of CP, and the same alteration is found in the taste buds of STZ-induced DM rats, whose sweet taste sensitivity is enhanced, at least in part, by decrease of their recognition threshold for sweet taste due to the increased expression of sweet taste receptors. Therefore, it is suggested that the blood glucose level is a regulator of sweet taste sensitivity by controlling the sweet taste receptor expression in taste buds of CP.

## Figures and Tables

**Figure 1 nutrients-12-00990-f001:**
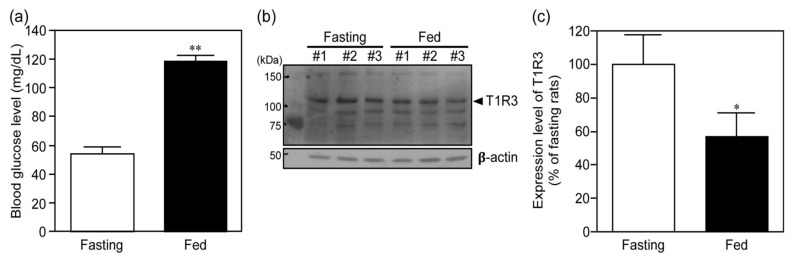
Blood glucose levels and expression of T1R3 in the pancreas in rats under fasting and fed conditions. After 18 h-fasting with or without subsequent 4 h-fed treatment of rats, their blood glucose levels (**a**) and expression of T1R3 in the pancreas (**b**,**c**) were measured. The results shown in panel **c** were obtained based on the band density of T1R3 corrected for by that of ß-actin shown in panel **b**. In panel **b**, #1, #2 and #3 are the rat numbers. Each bar represents the mean ± SD for three rats. * *p* < 0.05 and ** *p* < 0.001 (vs. fasting).

**Figure 2 nutrients-12-00990-f002:**
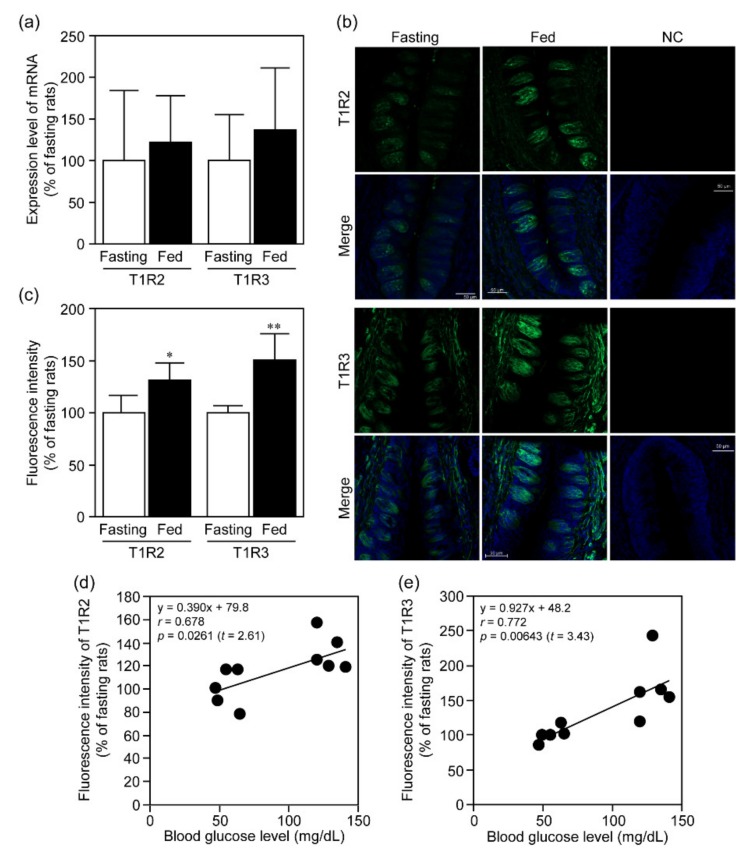
Expression of T1R2 and T1R3 in rat CP under fasting and fed conditions. After 18 h-fasting with or without subsequent 4 h-fed treatment of rats, the expression levels of T1R2 and T1R3 in CP isolated from their tongues were determined by real-time PCR (**a**) and immunohistochemical analysis (**b**,**c**). In panel **a**, mRNA expression levels of T1R2 and T1R3 were normalized against the corresponding levels of β-actin mRNA. In panel **b**, green and blue signals indicate expression of T1R2 or T1R3 and nuclei, respectively, and the results for fluorescence intensity derived from the immunoreactivity of T1R2 and T1R3 are given in panel **c**. In panel **b**, we linearly adjusted the contrast of whole images including the background by changing the input value from 255 to 200 in the images for T1R2 in fasting and fed groups. NC (negative control) was prepared by omitting the primary antibodies. Bar = 50 µm. Each bar represents the mean ± SD for 4 (**a**), and 4 (fasting of T1R3) or 5 (fasting and fed T1R2 and fed T1R3) (**b**,**c**) rats. **p*<0.05 and ***p*<0.01 (vs. fasting). Panels **d** and **e** indicate correlation between blood glucose levels and expression levels of T1R2 or T1R3 in rats.

**Figure 3 nutrients-12-00990-f003:**
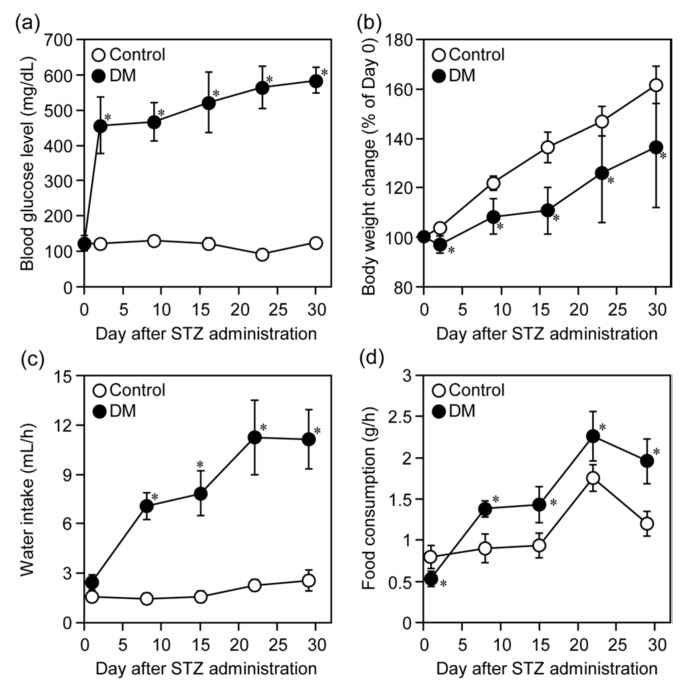
STZ administration induced DM conditions in rats. Blood glucose levels (**a**), body weight (**b**), water intake (**c**), and food consumption (**d**) in rats were measured just before (day 0) and weekly for 3 weeks after *i.p.* administration of STZ (50 mg/kg) to them. Each point represents the mean ± SD for 10 (control) and 6 (STZ) rats. * *p* < 0.001 (vs. control).

**Figure 4 nutrients-12-00990-f004:**
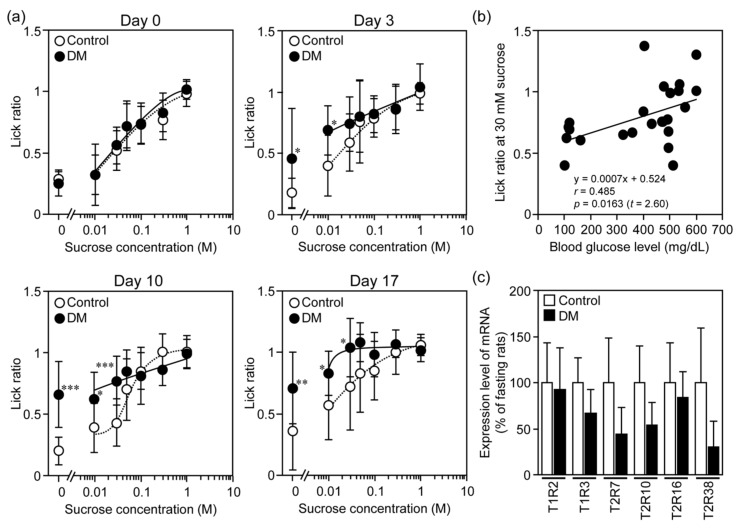
Alteration of the sensitivity and expression of receptors for sweet and bitter taste in STZ-induced DM rats. (**a**) Lick ratios were measured by means of a brief access test just before (day 0), and 3, 10 and 17 after *i.p.* administration of STZ (50 mg/kg) to them. Panel **b** indicates correlation between blood glucose levels and lick ratios at 30 mM sucrose on Days 0, 3, 10 and 17 in rats. (**c**) The expression levels of mRNAs for sweet (T1R2 and T1R3) and bitter (T2R7, T2R10, T2R16 and T2R38) taste receptors in CP isolated from their tongues on day 35 were determined by real-time PCR. mRNA expression levels of taste receptors were normalized against the corresponding levels of β-actin mRNA. Each point/bar represents the mean ± SD for 10 (control) and 6 (STZ) rats. * *p* < 0.05, ** *p* < 0.01, *** *p* < 0.001 (vs. control).

**Figure 5 nutrients-12-00990-f005:**
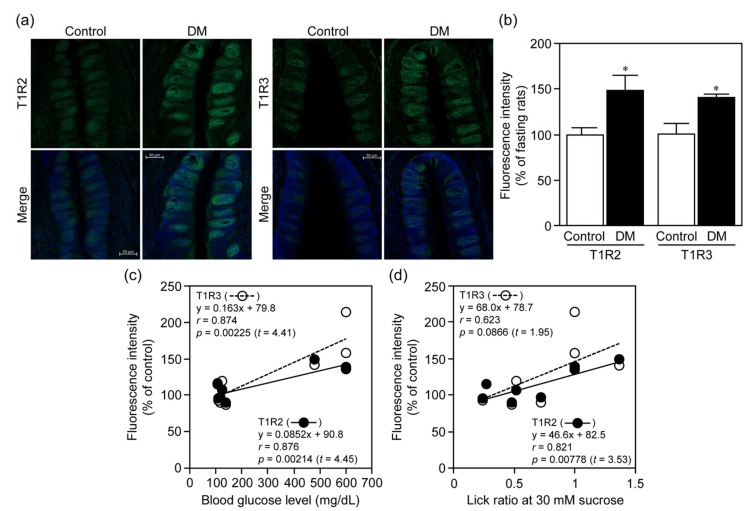
Protein expression of sweet taste receptors in CP of STZ-induced DM rats. The expression levels of T1R2 and T1R3 in CP isolated from the tongues of rats on day 35 were determined by immunohistochemical analysis. In panel **a**, green and blue signals indicate expression of T1R2 or T1R3 and nuclei, respectively, and the results for fluorescence intensity derived from the immunoreactivity of T1R2 and T1R3 are given in panel **b**. In panel **a**, we linearly adjusted the contrast of whole images including background by changing the input value from 255 to 150 in the images for T1R2 and T1R3 in control and DM groups. Bar = 50 µm. Each bar represents the mean ± SD for 5 (control) or 3 (STZ) rats. * *p* < 0.01 (vs. control). Panels **c** and **d** indicate correlation between blood glucose levels and expression levels of T1R2 or T1R3 on Day 35, and between lick ratios at 30 mM sucrose on Day 17 and expression levels of T1R2 or T1R3 on Day 35, respectively, in rats.

**Table 1 nutrients-12-00990-t001:** Primer sets used for quantitative real-time PCR.

Genes	Direction	Sequence	Size (bp)	Accession No.
*Tas1r2* (T1r2)	Forward	5′-TTCTCATGCTTCTGCCGACAG-3′	105	NM_001271266
	Reverse	5′-GCCAATCTTGAAGACACACACGA-3′		
*Tas1r3* (T1r3)	Forward	5′-AACAACCAATGGCTCACCTCC-3′	202	NM_130818.1
	Reverse	5′-AAAGCCATCAAGTACCAGGCAC-3′		
*Tas2r121* (T2R7)	Forward	5′-ACTCTATGCCACTTACTTCATATCC-3′	120	NM_023997.1
	Reverse	5′-AATGAGTGGCTTGAAGGGTAG-3′		
*Tas2r110* (T2R10)	Forward	5′-GGTCAATGCCAAAGGACCC-3′	301	NM_001166677.1
	Reverse	5′-TTAGGGATCCATGATGTGTATATGC-3′		
*Tas2r108* (T2R16)	Forward	5′-ATTCCATATTCAATCGCTGCC-3′	201	NM_001024686.1
	Reverse	5′-TCAGTTACTAACGAAATCCCGC-3′		
*Tas2r138* (T2R38)	Forward	5′-TATGTGGTGTCATTCTGTGCC-3′	207	NM_001024685.1
	Reverse	5′-GACTCTTCTCACCTTTTGCCT-3′		
*bactin* (β-actin)	Forward	5′-TGACCCTGAAGTACCCCATTG-3′	81	NM_031144.3
	Reverse	5′-TGTAGAAAGTGTGGTGCCAAATC-3′		

**Table 2 nutrients-12-00990-t002:** Antibodies used for Western blotting.

Proteins	Primary Antibody	Secondary Antibody
T1R3	rabbit anti-T1R3, dilution: 1:1000 (Cat No.: OST00259, Osenses, Keswick, SA, Australia) Blocking buffer: 3% BSA in PBS-T	anti-rabbit HRP-linked IgG, dilution: 1:10000 (Cat No.: PI-1000, Vector Laboratories, Burlingame, CA,USA)
ß-actin	rabbit anti-beta actin, dilution: 1:1000 (Cat No.: GTX109639, GeneTex Int. Co., Los Angeles, CA, USA) Blocking buffer: 4% BlockAce^®^ (Cat No.: UK-B40, KAC Co. Ltd., Hyogo, Japan) in PBS-T	

**Table 3 nutrients-12-00990-t003:** Antibodies used for immunohistochemistry.

Proteins	Primary Antibody	Secondary Antibody
T1R2	rabbit anti-TAS1R2, dilution: 1:200 (Cat No.: NB110-74920-azide, Novus Biologicals, LLC., Centennial, CO)	Donkey anti-rabbit IgG conjugated with Alexa Fluor 488, dilution: 1:1000 (Cat. No.: 21206, Life Technologies, Carlsbad, CA, USA)
T1R3	rabbit anti-T1R3, dilution: 1:1000 (Cat No.: OST00259, Osenses)	

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
