# Peer review of "Elevation of the Blood Glucose Level is Involved in an Increase in Expression of Sweet Taste Receptors in Taste Buds of Rat Circumvallate Papillae"

_nutrients, 2020, doi:10.3390/nu12040990_

Round 1

Reviewer 1 Report

N/A. 

Reviewer 2 Report

The authors have addressed all of my comments to my satisfaction.  A minor point is that the added text focused on the orosensory (rather than post-ingestive) basis of brief-access testing should be cited.  There are several excellent reviews by Spector and colleagues on this topic.

This manuscript is a resubmission of an earlier submission. The following is a list of the peer review reports and author responses from that submission.

Round 1

Reviewer 1 Report

The authors hypothesize that elevated circulating glucose affects the sweet taste signaling machinery in rat taste buds leading to altered behavioral sensitivity to sweet stimuli.  Two main models are tested, in which glucose is either acutely increased by post-fasting food intake, or chronically elevated by an injection of streptozotocin (STZ) in a rat model of diabetes.  This is an area of great interest in the field as investigators seek mechanisms linking obesity, diabetes, and taste. However, the behavioral experimental design complicates the interpretation of the results and needs careful, clearer explanation.  The same is true of sweet taste receptor analyses and prior studies referenced as detailed below. 

Major comments:

Introduction, p. 2, lines 44-52. The authors reference non-human primate studies to argue that taste buds, not taste nerve signals and central taste activity, are regulated by satiety.  It would be more straightforward to point out that the regulation of taste buds by satiety-related hormones is supported by many rodent studies.  Also, brief-access testing is suitable for testing orosensory responses without a significant post-ingestive contribution. 

The authors measured licking to sucrose mixed in 0.3 mM quinine in brief-access testing. The rationale for this choice is given in the Methods (2.5 Brief access test, p. 12) as to: “allow sensitive detection of a change in sweet taste sensitivity, because a rat, which can not detect the sweet tastant in the mixture, only perceives the bitter taste and thus avoids drinking the mixture”.  First, if the rat is unable to detect the sweet stimulus, how will this design allow them to test sweet taste sensitivity?  Second, a paper by Ninomiya and colleagues is cited in which rats are given the sweet taste receptor antagonist, gurmarin, prior to behavioral testing.  This antagonist was not part of the study design here.  Please clarify and add that a mixture of sweet+bitter was used in behavioral tests to Abstract and Introduction.

The authors conclude that licking responses to very low concentrations of sucrose in STZ-treated vs. control rats could differ because of reduced sweet or enhanced bitter sensitivity (Fig. 4a, Discussion). Non-significant decreases in bitter taste receptor mRNA at day 35 after STZ (Fig. 4b) are used to support this explanation.  This is a risky argument with P values of around 0.08-0.10. 

Sweet taste receptor protein level, but not mRNA, is significantly altered by acute or chronic elevation of blood glucose. It seems that the authors’ explanation for this uncoupling is that feeding causes an immediate cellular response.  What then is the reason for unchanged gene expression in diabetic rats at day 35?  Please clarify. 

Minor comments

Materials and Methods

3 Fasting and fed conditions, p. 2. Were fasted rats allowed access to water? 5 Brief access test, p. 3. What is lick volume? 6 Exfoliation of epithelial tissue…, p. 3. Specify injection route of enzyme cocktail. 7 qRT-PCR, p. 4. Specify method used to analyze/normalize mRNA expression rather than just % as on figure legends.

Results

Figure 2 legend. Were experimental replicates used to calculate means and if so, how many? Figure 5 legend. T1R2 and T1R3 expression appears dimmer than in Fig. 2.  Please note if that is indeed true, since intensity measurements are normalized.  Also note if adjustment of brightness and contrast differed in the two images. Report interlick intervals.

Discussion

9, line 285, line 322. …”STZ-treated rats, who had no insulin secretion”.  Report insulin values if tested or rephrase and reference if not tested. 9, line 288. “This is almost identical to the finding of Zverev that the recognition threshold for sucrose in human subjects was higher 1 hr after a meal…(italics added)”.  Rewrite this section to reflect that (1) humans vs. rats; (2) psychophysics vs. licking; (3) sucrose vs. sucrose+quinine; (4) 4-hr vs. 1-hr.  9, line 298-299. In the findings of Mace et al., did T1R3 mRNA change?

Conclusions

Clarify that sweet receptor protein was increased and that licking responses to low sucrose concentrations in a sweet+bitter mixture

Author Response

Thank you for your very careful review and giving critical and fruitful comments. We modified our manuscript extensively following to your comments.

Major comments
Comment #1: Introduction, p. 2, lines 44-52. The authors reference non-human primate studies to argue that taste buds, not taste nerve signals and central taste activity, are regulated by satiety.  It would be more straightforward to point out that the regulation of taste buds by satiety-related hormones is supported by many rodent studies.  Also, brief-access testing is suitable for testing orosensory responses without a significant post-ingestive contribution.

Response #1: Following to the suggestion, we added the description on hormonal regulation of taste cell sensitivity in P. 2, lines 49-50, and on application of a brief access test as a suitable method for our objective in this study in P. 3, lines 101-103.

Comment #2: The authors measured licking to sucrose mixed in 0.3 mM quinine in brief-access testing. The rationale for this choice is given in the Methods (2.5 Brief access test, p. 12) as to: “allow sensitive detection of a change in sweet taste sensitivity, because a rat, which can not detect the sweet tastant in the mixture, only perceives the bitter taste and thus avoids drinking the mixture”.  First, if the rat is unable to detect the sweet stimulus, how will this design allow them to test sweet taste sensitivity?  Second, a paper by Ninomiya and colleagues is cited in which rats are given the sweet taste receptor antagonist, gurmarin, prior to behavioral testing.  This antagonist was not part of the study design here.  Please clarify and add that a mixture of sweet+bitter was used in behavioral tests to Abstract and Introduction.

Response #2: The description on the method of the brief access test in the previous version of the manuscript might lead misunderstanding of readers. Thus, we modified the description in P. 3, lines 122-123, and added the description on use of a brief access test with a mixture of sucrose and QHCl in P. 1, lines 20-21 (Abstract) and P. 2, lines 62-63 (Introduction).

As for gurmarin, a sweet taste receptor antagonist, the paper of Ninomiya and his collaborators cited was to evaluate the effect of gurmarin on sweet taste sensitivity in mice, and for sensitive detection of alteration of the sensitivity of mice, they adopted a brief access test with a mixture of sweet and bitter tastants. In this study, in contrast, we wanted to evaluate alteration of sweet taste sensitivity of mice, and thus we did not use gurmarin.

Comment #3: The authors conclude that licking responses to very low concentrations of sucrose in STZ-treated vs. control rats could differ because of reduced sweet or enhanced bitter sensitivity (Fig. 4a, Discussion). Non-significant decreases in bitter taste receptor mRNA at day 35 after STZ (Fig. 4b) are used to support this explanation.  This is a risky argument with P values of around 0.08-0.10.

Response #3: We agree with your opinion on the expression of bitter taste receptors. Thus, we changed the phrase on it (with a tendency of a decrease of mRNA expression) in P. 11, lines 33--342 in this version of the manuscript.

Comment #4: Sweet taste receptor protein level, but not mRNA, is significantly altered by acute or chronic elevation of blood glucose. It seems that the authors’ explanation for this uncoupling is that feeding causes an immediate cellular response.  What then is the reason for unchanged gene expression in diabetic rats at day 35?  Please clarify.

Response #4: We have no reasonable explanation on mechanism underlying increased expression of sweet taste receptors in STZ-induced DM rats. However, we think a possibility that the post-transcriptional regulation of sweet taste receptor expression in fasting and fed rats might become chronic in the DM ones. The description on this was given in P. 12, lines 372-375.

Minor comments
Comment #1: Materials and Methods

Comment #1-1: 3 Fasting and fed conditions, p. 2. Were fasted rats allowed access to water?

Response #1-1- We added the sentence on the free access to water in P. 2, lines 88-89.

Comment #1-2: 5 Brief access test, p. 3. What is lick volume?

Response #1-2: We added the phrase on mean of "lick number" in P. 3, lines 105-106.

Comment #1-3: 6 Exfoliation of epithelial tissue…, p. 3. Specify injection route of enzyme cocktail.

Response #1-3: We added the word "subcutaneous" in P. 4, lines 144.

Comment #1-4: 7 qRT-PCR, p. 4. Specify method used to analyze/normalize mRNA expression rather than just % as on figure legends.

Response #1-4: We added the description on the normalization of mRNA expression levels in P. 4, lines 156-157, in the legends for Figure 2 in P. 7, lines 239-240, and Figure 4 in P. 9, lines 279-280.

Comment #2- Results

Comment #2-1: Figure 2 legend. Were experimental replicates used to calculate means and if so, how many?

Response #2-1: As experimental replicates in Fig. 2, we have indicated the number of rats used in experiments P. 7, lines 245-246.

Comment #2-2: Figure 5 legend. T1R2 and T1R3 expression appears dimmer than in Fig. 2.  Please note if that is indeed true, since intensity measurements are normalized.  Also note if adjustment of brightness and contrast differed in the two images.

Response #2-2: Following your suggestion, we replaced the images in Figs. 2 and 5, and to obtain clear and comparable immunofluorescence in independent experiments, we linearly adjusted the contrast of whole images including background by changing the input value from 255 to 200 in the images for T1R2 in fasting and fed groups in Fig. 2b, and that from 255 to 150 in the images for T1R2 and T1R3 in control and DM groups in Fig. 5a, using the Photoshop software. The description on this was given in Materials and Methods in P. 5, lines 186-190, and in figure legends in Fig. 2 in P. 7, lines 242-244 and in Fig. 5 in P. 9, lines 303-305.

Comment #2-3: Report interlick intervals.

Response #2-3: We set 10 sec-presentation of distilled water as the interval between the two taste solutions as indicated in Materials and Methods in P. 3, lines 132-134.

Comment #3- Discussion

Comment #3-1: 9, line 285, line 322. …”STZ-treated rats, who had no insulin secretion”.  Report insulin values if tested or rephrase and reference if not tested.

Response #3-1: We added the two references of Nos. 25 and 26 on no insulin secretion in STZ-treated rats in P. 11, line 323, and in References in P. 14, lines 452-455.

Comment #3-2: 9, line 288. “This is almost identical to the finding of Zverev that the recognition threshold for sucrose in human subjects was higher 1 hr after a meal…(italics added)”. Rewrite this section to reflect that (1) humans vs. rats; (2) psychophysics vs. licking; (3) sucrose vs. sucrose+quinine; (4) 4-hr vs. 1-hr.

Response #3-2: Following your suggestion, we extensively modified the sections from P. 11, line 320 to P. 12, line 375.

Comment #3-3: 9, line 298-299. In the findings of Mace et al., did T1R3 mRNA change?

Response #3-3: Mace et al. did not evaluate mRNA levels of T1R3 in their study.

Comment #4- Conclusions

Clarify that sweet receptor protein was increased and that licking responses to low sucrose concentrations in a sweet+bitter mixture.

Response #4: We think the increase of sweet taste sensitivity in STZ-induced DM rats is caused by decrease of their threshold for detection of sweet tastants due to the increased expression of sweet taste receptors. The description on this was added in P. 12, lines 383-384.

Reviewer 2 Report

“Elevation of the Blood Glucose Level Is a Trigger of an Increase in Expression of Sweet Taste Receptors in Taste Buds of Rat Circumvallate Papillae” The authors They examined sweet taste receptors T1R2 and T1R3 in circumvallate papillae (CP) using fast/fed rat model and STZ-induced diabetic model. In conclusion, they found elevation of blood glucose level was involved in alteration of the expression of sweet taste receptors in CP. The manuscript is more about phenotypic observation linking sweet taste receptors expression with blood glucose. However, the authors did not show direct evidence to prove that blood glucose level trigger an increase in sweet taste receptors.

Comments:

1b. It does not look like T1R3 protein level is reduced in Fed state base upon the blots. 2b. Antibodies are not for rat. Have to double check the fluorescent signals are not autofluorescence. 4. The results in the manuscript showed STZ increases lick ratio starting from Day 3. It is better to have a correlation of blood glucose level and lick ratio as well as sweet taste receptors protein level. 5. It is better to quantify the immunoblot results of sweet taste receptors. To link sweet taste receptors to elevated blood glucose, it is better to investigate sweet taste receptors knockdown cells or knockout animal models.

Author Response

Thanks for your very careful reviewing and giving critical and fruitful suggestions. We added new data and modified our manuscript extensively.

Comment #1: 1b. It does not look like T1R3 protein level is reduced in Fed state based upon the blots.

Response #1: Following to your pointing out, in Fig. 1b, we replaced the Western blot images by which the quantitative results were reflected more clearly.

Comment #2: 2b. Antibodies are not for rat. Have to double check the fluorescent signals are not autofluorescence.

Response #2: We added the photomicrographs of negative controls for T1R2 and T1R3 in Fig. 2, to demonstrate the fluorescence obtained is derived from specific immunoreactivity of the antibodies used. The legends on this was added to Figure 2 in P. 7, lines 244-245.

Comment #3: 4. The results in the manuscript showed STZ increases lick ratio starting from Day 3. It is better to have a correlation of blood glucose level and lick ratio as well as sweet taste receptors protein level.

Response #3: Thanks for your critical suggestion to strengthen our conclusion. We added the graphs for correlation between blood glucose levels and expression of T1R2 and T1R3 in fasting and fed rats in Fig. 2d and 2e, between blood glucose levels and lick ratios in DM rats in Fig. 4b, between blood glucose levels and expression levels of T1R2 and T1R3 in Fig. 5c, and between lick ratios and expression levels of T1R2 and T1R3 in Fig. 5d. The legends on this was added to Figure 2 in P. 7, lines 246-248, Figure 4 in P. 9, lines 275-277, and Figure 5 in P. 9-10 lines 306-309. The description on them were added in P. 6, lines 211-212, P. 7, lines 254-257, P. 9, lines 291-297, and P. 10, lines 316-317.

Comment #4: 5. It is better to quantify the immunoblot results of sweet taste receptors. To link sweet taste receptors to elevated blood glucose, it is better to investigate sweet taste receptors knockdown cells or knockout animal models.

Response #4: We absolutely agree to your opinion. We tried immunoblots for T1R2 and T1R3 in rat taste buds, but we just obtained no or faint bands for them and this seemed to be due to small sample amounts. Thus, we adopted immunofluorescent quantification with immunohistochemistry. As for use of sweet taste receptor-knockout cells/animals, we will report on the findings with them in near future.

Round 2

Reviewer 2 Report

I know the author replaced the blots in Fig. 1b. I am still not convinced that TIR3 protein level is down in Fed state. Please provide an explanation.

Author Response

Comments- I know the author replaced the blots in Fig. 1b. I am still not convinced that TIR3 protein level is down in Fed state. Please provide an explanation.

Response- Thanks for your detail confirming. In the quantification of the blot, the band density of T1R3 in fed rats was almost the same with that in fasting ones, but the density of GAPDH was apparently greater in the former than in the latter, as shown in Fig. 1b. Thus, the relative expression level of T1R3 was less in fed rats than in fasting ones.